

# Identifying Snow-Covered Areas from Unoccupied Aerial Systems (UAS) Visible Imagery: A Comparison of Methods

Mahsa Moradi[1,2], Andrew G. Fleming[1,2], Adam Hunsaker[1,2], Jennifer M. Jacobs[1,2]

5   [1]Department of Civil and Environmental Engineering, University of New Hampshire, Durham, NH, USA

[2]Earth Systems Research Center, Institute for the Study of Earth, Oceans, and Space, University of New Hampshire, Durham, NH, USA

*Correspondence to:* Mahsa Moradi (mm1631@usnh.edu),





**Abstract**

High-resolution imagery from Unoccupied aerial systems (UAS) offers new opportunities for mapping snow cover at fine spatial scales, particularly in regions with ephemeral and variable snow conditions. This study evaluates a range of classification strategies for generating snow-covered area (SCA) maps from UAS imagery with 3 cm pixels collected over open areas in southern New Hampshire, USA and offers practical recommendations for producing UAS-derived SCA maps. We tested machine learning and threshold-based approaches, exploring the influence of input features and training set composition on classification accuracy and generalization. Results show that classifiers using full red-green-blue inputs, including Maximum Likelihood Estimation (MLE), Support Vector Machine (SVM), and Random Forest (RF), consistently yield high performance (accuracy, balanced accuracy and f1 score above 0.96) and transfer well across sensors and locations. In contrast, approaches relying solely on the blue band (including SVM, static and dynamic thresholding) exhibited lower balanced accuracy (0.83 to 0.86) and limited generalizability. Training with fewer than 12 orthomosaics reduced the reliability and consistency of snow cover classifiers. When fewer flights are possible, UAV flights that collectively capture the full range of snow-covered area (fSCA) between 20–60%, partial melt, and refrozen surfaces, should be prioritized. We conclude that highly accurate snow mapping from routine UAS optical imagery is possible even in landscapes with variable snow cover and ephemeral, shallow snowpacks.

**Keywords:** Unoccupied aerial systems (UAS), snow-covered area (SCA), Optical imagery, Machine Learning, Blue band thresholding



## 1 Introduction

The extent, duration, and patterns of snow-covered areas (SCA) and their accurate

delineation provide insight into the winter hydrologic cycle including snow accumulation,

redistribution and melt as indirect understanding of soil freeze-thaw state and evaporation

(Pomeroy et al. 2007; Dong, 2018; Moradi et al., 2023) and, when summarized over time,

characterize a region's snow climatology (Johnston et al., 2023). The states and process are

important for characterizing snow melt water runoff (Immerzeel et al., 2009; Hammond et al.,

2018; Le et al., 2023) as well as the exchange of heat and moisture between the land surface and

atmosphere (Walland and Simmons, 1996; Liston, 1999; Zhang, 2005; Lundquist et al., 2024) and

ecosystem controls (Liston, 1999; Baptist et al., 2010; Panchard et al., 2023).

Snow cover varies at multiple scales, and the appropriate scale depends on the application

(Blosch, 1999).  For many applications requiring continuous spatial coverage, satellite-scale

observations are suitable (Dong, 2018; Johnston et al., 2023). For other applications requiring fine

(<1 m) to moderate (10s of meters) spatial resolutions (e.g., Mott et al., 2017; Sproles et al., 2018;

Harder et al., 2019; Barna et al., 2020), snow presence or absence may deviate from the coarse

satellite scales and require invoking various physical features to downscale the satellite

observations (Clark et al., 2011; Sturm, 2015).

Unoccupied aerial systems (UAS) are emerging as a reliable, low-cost method for

collecting snow data at finer spatial resolutions than satellites and are not limited by orbital revisit

times (Sturm, 2015; Verfaillie et al., 2023). Platforms outfitted with electro-optical (EO) cameras,

which are becoming increasingly available, can provide detailed spatial information on demand in

cold regions and provide numerous research opportunities (Gaffey and Bhardwaj, 2020; Verfaillie

et al., 2023). While most snow cover mapping strategies have used either (1) a combination of the



visible and infrared, or (2) microwave, there is mounting evidence that multispectral imagery (i.e.,

red-green-blue (RGB) alone or RGB and NIR) can provide reasonable results. Additionally, UAS

snow cover performance does not appear impacted to the same extent as satellite snow cover

products in forested areas, complex terrain, and shallow snow conditions (<10 mm depth) (Ault et

al., 2006; Hall and Riggs, 2007; Gao et al., 2010; Frei et al., 2012).

In contrast to satellite approaches which typically use a Normalized Difference Snow Index

method (NDSI, Dozier, 1989) to map snow cover using green and SWIR bands, there is no single

recommended strategy for mapping SCA using UAS RGB imagery. Promising methods for

differentiating between snow and non-snow in UAS RGB imagery include machine learning (ML)

classifiers such as maximum likelihood, random forest, support vector machine, k-means

clustering (Liang et al., 2017, Belmonte et al., 2021, Johnston et al., 2025; Niedzielski et al., 2018),

and thresholding approaches such as static or dynamic thresholding of blue-channel thresholding

(Jacques-Hamilton et al., 2025; Eker et al., 2019). Each of these methods appears to be capable of

mapping SCA with different strengths and weaknesses. Multiple studies have used K-means

clustering to distinguish between bare ground and snow, as well as ice and shadows, but the method

requires manual, qualitative interpretation of the classes making it challenging to apply for

multiple images (Niedzielski et al., 2018; Johnston et al., 2025). Static and dynamic blue band

thresholding classifiers have shown promising results for SCA mapping, but results appear to be

dependent on lighting conditions, shadowing effects, and spectral features of existing objects (Eker

et al., 2019; Jacques-Hamilton et al., 2025). Additionally, there is no standard to determine the

optimum blue band threshold for an orthomosaic due to varying conditions. Eker et al. (2019)

determined the threshold as the minimum value of sample snow-covered pixels selected randomly

from different areas. Jacques-Hamilton et al. (2025) determined the threshold value by visually



inspecting snow-covered pixels and adjusting the threshold until a satisfactory alignment with the

underlying imagery was achieved. However, this visual-based approach is inherently subjective,

relies on human judgment, and is not scalable for automated application across large numbers of

orthomosaics.

Despite the promise of these tools, the different approaches have yet to be compared

directly. Even indirect comparisons are challenging because the previous studies report different

performance metrics. A systematic evaluation is necessary to understand the trade-offs between

threshold-based approaches and the improved classification performance alongside greater

training complexity and reliance on labeled data, offered by supervised machine learning

classifiers in the context of snow mapping. Beyond differences in underlying assumptions and

structure of previously used classifiers, the choice of predictor variables, whether limited to blue-

band intensity or using the full range of RGB values, require more investigation. Furthermore,

because the previous studies typically focused on the application of derived SCA products, there

is limited guidance on strategies for UAS data collection to support development of automated,

accurate and robust SCA mapping tool.

Here, our goal is to compare the performance of promising snow cover classification

approaches in relatively homogenous open areas and to identify one or more classifiers that can

consistently and accurately map SCA with limited manual input. We use 36 orthomosaics from

repeat-pass UAS visible imagery over two homogenous open areas in Southeastern New

Hampshire, USA. The data was collected over multiple years. First, manually classified, randomly

selected pixels are used to train and evaluate the performance of three supervised classifiers (i.e.,

Maximum Likelihood Estimation (MLE), Random Forest (RF), Support Vector Machine (SVM))

with the three visible bands (i.e., RGB) as well as blue-channel only methods via SVM, static and





dynamic thresholding. We examined three generalization scenarios relevant to practical snow mapping applications: (1) cross-sensor generalization, where a model trained on orthomosaics acquired with one UAS camera is applied to imagery from a different sensor; (2) cross-site

transferability, involving the use of a model trained at one location to classify data from another, similar site; and (3) training data sparsity, in which the number of orthomosaics used for training is reduced to evaluate the sensitivity of classifier performance to limited data availability. Finally, we synthesize our findings and provide recommendations aimed at addressing key operational constraints in UAS-based snow mapping. We emphasize the critical role of representative UAS

data collection as a prerequisite for developing robust and generalizable SCA classifiers.



## 2 Methods

### 2.1 Site Descriptions

Two field sites, the University of New Hampshire's Thompson Farm Research Observatory
(43.1089° N, 70.9485° W) and Kingman Farm Research Observatory (43.17207° N, 70.92662°
W), are located in southeastern New Hampshire, United States. The sites are separated by
approximately 8 km. Both farms are actively managed and operated by the University of New
Hampshire. Analyses were performed on the open field portions of the sites indicated by the
hatched areas in Fig. 1.

The Thompson Farm study field is approximately 7.6 ha with elevations ranging from 10
to 35 m. The site has limited topographic relief with gentle slopes where low-lying areas within
the field and clayey soils result in areas of surface water collection and icing during the winter
months (Jacobs et al., 2021). Field vegetation varies between low-cut pasture grass and tall (~1 m)
unkempt grasses. The surrounding mixed deciduous and coniferous forest canopy is 10 to 30 m
tall (Jacobs et al., 2021). Most of the field experiences two to five hours per day of shade with up
to eight hours per day at the forest-field boundary in the north portion of the field (Cho et al.,
2021). An access road, running south to north, is the high point between the two fields with the
eastern field being lower than the western field. Soils at Thompson Farm are classified as silt loam,
sandy loam, and loamy sand in the National Cooperative Soil Survey (NCSS).

The Kingman Farm site consists of two prominent field sections, the main field and a
sheltered field, that were the focus of this study. The main field is approximately 17 ha with gentle
rolling terrain and maximum elevation changes of ~10 m. This field section is largely exposed
with a mixed forest (deciduous/coniferous) bordering the northern edge. An access road runs east
to west with the terrain on the southern side of the field. Field vegetation is primarily low-cut



pasture grass with a few small sections of tall (~1 m) unkempt grass along the access road. The

main field soil is classified as fine sandy loam, silt loam and loamy fine sand. The sheltered field

section at Kingman Farm is a smaller (~0.7 hectare) managed field surrounded by forest. The

northern edge of the field is bordered by tall conifers (~30 m), while the southern edge is bordered

by deciduous trees and an access road. The sheltered field is relatively flat with elevation variations

of approximately 5 m with the highest elevations near the center of the field. NCSS classifies soil

in this field as fine sandy loam.

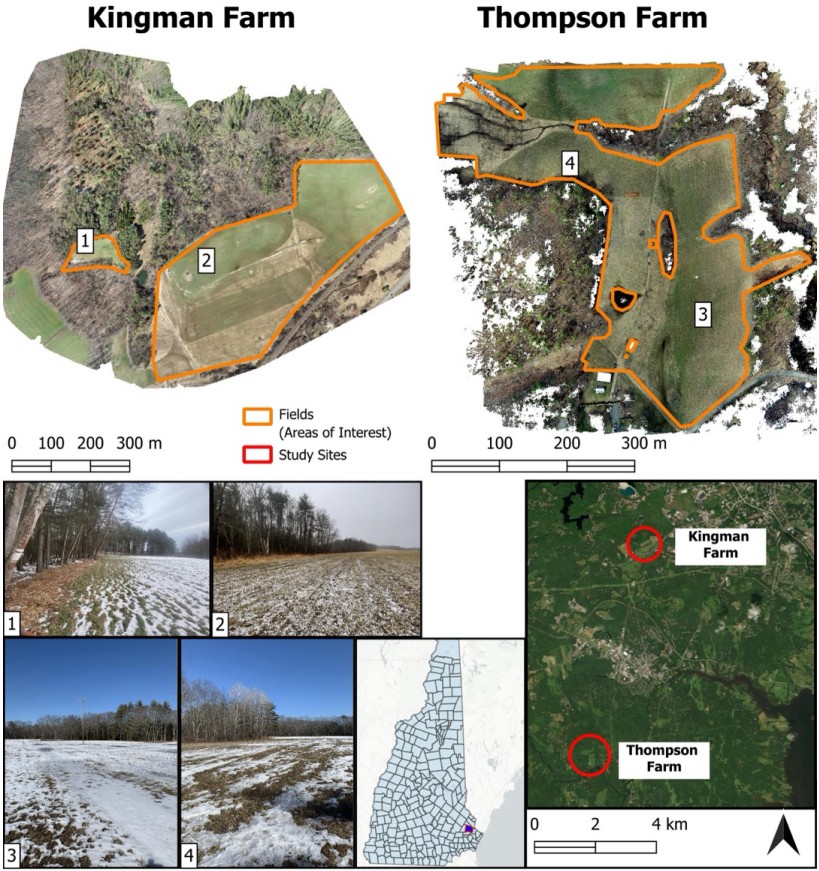

**Figure 1.** The two study sites, Kingman Farm and Thompson Farm, located at southern New Hampshire.




The Thompson Farm and Kingman Farm sites are located within the ephemeral snow class, which is characterized by relatively warm temperatures, shallow snow, and a total duration of snow cover lasting for an average of 70 days per year (Sturm and Liston, 2021; Johnston et al., 2023). Based on a 20-year (2002–2021) winter climatology at the co-located U.S. Climate Reference

Network sites (USCRN, Thompson site ID: 1040, Kingman Site ID: 1041), January is the coldest month with an average temperature of -4.1°C followed by February (-3.1°C) and December (-1.2°C). These are the only months having average temperatures below 0°C. From December to February, monthly mean high temperatures range from 0.6 to 3.4°C with lows ranging from -9.3 to -6.0°C. The average monthly liquid equivalent precipitation is highest in December (122 mm)

and lowest in January (78 mm). February, March, and April have approximately 100 mm of precipitation on average (98 to 108 mm). The sites' combination of sub-freezing temperatures and regular precipitation results in frequent snowfall. Conversely, because above-freezing temperatures are also common, there are intermittent melting and rain-on-snow events throughout the winter. Based on MODIS observations, snow cover occurs in November (average monthly

fraction snow covered area is 0.02), December (0.36), January (0.70), February (0.73), March (0.48), and April (0.04).

## 2.2 Data

### 2.2.1 UAS Optical Imagery

Three different UAS platforms were used throughout this study to collect RGB imagery.
The three systems are (1) the Phantom 4 RTK, (2) the Green Valley LiAir v70 flown on the DJI





M300, and (3) the Phoenix miniRanger3 flown on the DJI M50. Both the LiAirv70 and miniRanger3 have integrated lidar sensors that were used separately to derive snow depth products.

The DJI Phantom 4 RTK (real-time kinematic) has a 20-megapixel 1 inch CMOS optical sensor with a mechanical shutter. The exposure of each image was automatically adjusted using the average light metering mode with no exposure bias. In low light conditions, the Phantom 4 favors ISO and aperture adjustments over decreasing shutter speed to minimize the potential for blurring while collecting images during flight. All Phantom 4 flights were conducted at an altitude of 100 m, resulting in a ground sampling distance (GSD) of ~3 cm. Flight line spacing and flight speed were adjusted to achieve 70% side and front image overlap. Flight planning and image capture configuration were done using the DJI Pilot application built into the DJI smart controller. A digital elevation model (DEM) was uploaded and referenced to ensure a consistent flight altitude was maintained throughout each flight.

The Greenvalley LiAir v70 and the Phoenix miniRanger3 both have an integrated Sony A5100 camera as part of the sensor payload. The Sony A5100 has a 23.5 x 15.6 mm back illuminated APS-C CMOS sensor containing 24 megapixels. During operation, a fixed shutter speed of 1/1000s and a fixed aperture setting of f/5.6 were used. Automatic exposure adjustments were made solely by changing the ISO setting. The pattern light metering mode with no exposure bias was used to determine the appropriate ISO adjustment for each image capture. The fixed high shutter speed and large aperture opening prevents image blurring. Because both of these UAS are primarily lidar payloads, image overlap capture is not as easily customizable as with the Phantom 4 RTK. For this study, images captured with the v70/miniRanger3 were collected at a time interval of 2 seconds, resulting in a forward overlap of ~80%. Likewise, flightline spacing was chosen to



ensure 70% lidar swath overlap, resulting in ~80% image side overlap. Flightlines were made

using the UgCS flight planning software. A DEM was referenced within UgCS to ensure a

consistent flight altitude was maintained throughout each flight. Flights with the v70 were

conducted at an altitude of 50 m, whereas miniRanger3 flights were conducted at 80 m. The

resulting GSD was ~2 cm and ~3 cm, respectively.

For all surveys, the collected UAS imagery were stitched together using Agisoft Metashape

Pro photogrammetry software to produce RGB orthomosaics. The 3-band Red-Green-Blue (RGB)

orthomosaics were the primary input to the snow cover detection algorithms. The standard

photogrammetry workflow implemented in Metashape involves the generation of a surface model,

leveraging image overlap and varying image perspectives, followed by the projection of each

image onto the produced surface model. This process removes distortions (perspective/lens) from

each image and results in a true planimetric map of the survey area (see appendix for additional

photogrammetry processing details). Although positional accuracy of RGB imagery was not

critical for this study, all collected UAS image geotags were either post-processing kinematic

(PPK) or real-time kinematic (RTK) corrected prior to photogrammetry processing (see appendix

for further details).

A total of 36 UAS RGB orthomosaics were produced and used for this snow cover

detection study. Table 1 lists the number of orthomosaics produced each winter season by study

site and camera. UAS RGB imagery were collected at Thompson and Kingman Farm over the

course of five winter seasons, 2020-2025. Flights were scheduled to capture a range of snow

conditions. However, due to limited tie-points, most images collected during complete snow cover

could not be effectively stitched together and were excluded from this study. Of the 36 dates, 10





represented very shallow/patchy snow conditions with observed snow depths less than 2 cm, 4

were completely snow free, and the remaining 22 dates comprised of a more uniform snowpack.

Figures 2 and 3 show RGB orthomosaics acquired with the Phantom 4 at Kingman Farm (N = 15)

and Thompson Farm (N = 11), respectively. Figure 4 shows the RGB orthomosaics acquired at

Kingman Farm (N = 5) and Thompson Farm (N = 5) with the Sony A5100 camera.

**2.2.2 Ancillary Observations**

Snow depth was available on dates coinciding with UAS orthomosaics. On selected dates,

UAS lidar was used to produce field-scale high-resolution snow depth estimates. On dates when

lidar was not flown, snow depth observations were acquired from either automated ultrasonic snow

depth sensors (Judd Communications LLC) or from field camera time lapse images of snow stakes,

both deployed in the open field at the respective site. Field cameras were installed following the

method used in NASA's 2020 SnowEx field camera campaign in Grand Mesa, CO (Chris

Heimstra, personal communication, 16 November 2020). The observed snow depths (either UAS

lidar and/or in-situ) coinciding with the 36 UAS optical surveys had a mean of ~10 cm and a

standard deviation of ~ 9 cm.

Subhourly air temperature and shortwave radiation data was obtained from the USCRN

stations located at the study sites, through NOAA's publicly available archive. Air temperature

(°C) and incoming shortwave radiation (W/m²) from 2020 to 2025 at five-minute temporal

resolution were used to calculate the mean and standard deviation during each survey. Air

temperature at the start of each UAS survey, along with the mean air temperature over the six hours

preceding each flight, was used as a proxy to estimate snow wetness conditions. If the 6-hour mean

air temperature exceeded 0 °C, the snow was assumed to be wet at the start of the survey.



Additionally, the mean observed shortwave radiation during each flight was calculated to approximate sky conditions, following the method described in the Supplementary Material.

**Table 1.** Number of UAS surveys conducted during each winter season by field site (TF: Thompson Farm, KF: Kingman Farm) and sensor (P4: Phantom4, A5100: Sony A5100).

| Winter | Total Images | Field Site | | Sensor | |
|---|---|---|---|---|---|
| | | TF | KF | P4 | A5100 |
| 2020-2021 | 13 | 13 | 0 | 13 | 0 |
| 2021-2022 | 5 | 5 | 0 | 2 | 3 |
| 2022-2023 | 2 | 2 | 0 | 0 | 2 |
| 2023-2024 | 9 | 0 | 9 | 9 | 0 |
| 2024-2025 | 7 | 0 | 7 | 2 | 5 |
| Total | 36 | 20 | 16 | 26 | 10 |





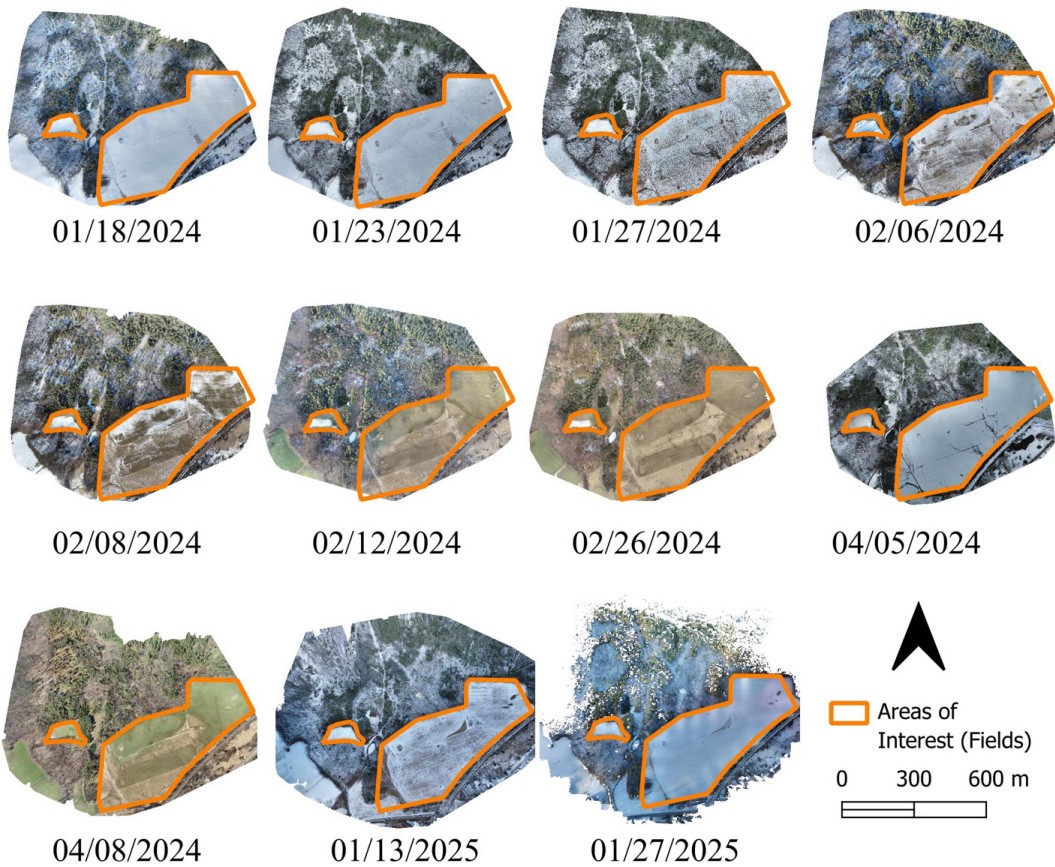

**Figure 2.** RGB orthomosaics of Kingman Farm acquired with the Phantom 4. Survey dates are shown below each orthomosaic. Areas of interest are outlined in orange.



12/23/2020   01/25/2021   02/20/2021   02/23/2021

02/24/2021   02/26/2021   02/28/2021   03/01/2021

03/03/2021   03/07/2021   03/10/2021   03/11/2021

04/02/2021   01/12/2022   01/26/2022

Area of Interest
(Field)

0   150   300 m

240

**Figure 3.** RGB orthomosaics of Thompson Farm acquired with the Phantom 4. Survey dates are shown below each orthomosaic. Areas of interest are outlined in orange.



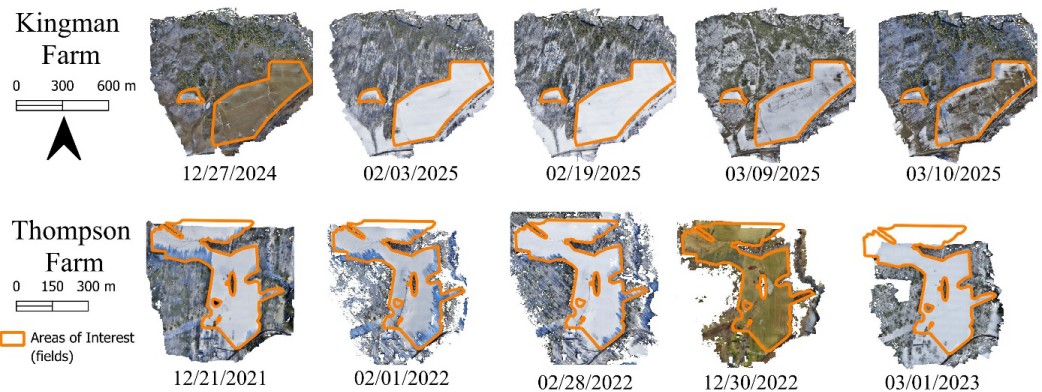

**Figure 4.** RGB orthomosaics of Kingman Farm (top row) and Thompson Farm (bottom row) acquired with the Sony A5100 camera. Survey dates are shown below each orthomosaic. Areas of interest are outlined in orange.

### 2.2.3 Ground Truth Dataset for Snow Cover Classification

For each orthomosaic, the field area of interest process was isolated, and the surrounding forested areas was removed using site-specific masks. Over-saturated pixels (e.g., [255, 255, 255] or [0, 0, 0]) were also removed prior to analysis. The fields of each orthomosaic were delineated in QGIS and 50 points were randomly selected within these areas of interest. A new set of random points were generated for every orthomosaic. For Kingman Farm, 48 points were placed in the open pasture and two in the sheltered field to represent the relative areas of the two fields. The pixel nearest to each sample point was classified manually as snow-covered or bare ground. The red, green, and blue band values for each of the randomly sampled pixel were extracted.

### 2.3 Snow Cover Classifiers

This study employed five classifiers for snow cover mapping, including both machine learning approaches (Random Forest (RF), Support Vector Machine (SVM), and Maximum Likelihood Estimation (MLE)) and thresholding approaches (static and dynamic). The choice of



predictors varied by classifier, utilizing either full RGB intensity values or exclusively the blue band. The combination resulted in seven distinct classification configurations. All classifiers were implemented in Python using scikit-learn library. When hyperparameter tuning was required, it

was performed using grid search. Implementation details for each classifier are provided below.

The random forest was developed using RGB color intensity values as predictive features. The search space for hyperparameters included variations in the number of estimators, maximum tree depth, minimum samples required to split an internal node, and the minimum number of samples required to be at a leaf node. To identify the optimal set of hyperparameters, 5-fold cross-

validation was employed. The set of parameters yielding the highest mean balanced accuracy across the cross-validation folds was selected as the optimal set.

Two variations of the SVM classifier were applied using RGB intensity values and only blue band intensity values (SVM Blue Band). Hyperparameter tuning was performed to identify the optimal combination of the regularization parameter (C), kernel type (linear, radial basis

function, polynomial, or sigmoid), and kernel coefficient (gamma). 5-fold cross-validation, along with balanced accuracy as the evaluation criterion, was used to determine the optimal hyperparameters.

Modeled off the Maximum Likelihood Classification tool in ENVI (Environment for Visualizing Images) version 6.1, a MLE classifier was implemented in python to predict snow

cover. This classification method assumes that the band values are normally distributed. Under this assumption, class statistics (i.e. mean vectors, covariance matrices, and prior probabilities) were computed from the training set for each class. The discriminant function determined the probability of each sample belonging to each class and the sample was assigned to the class with the highest probability.



The static blue band thresholding classifier applied a single, fixed threshold to the blue band intensity values. A pixel was classified as snow-covered if its blue band intensity exceeded this threshold. The optimal threshold selection was guided by 5-fold cross-validation on a range of candidate thresholds, using balanced accuracy (i.e., the average accuracy of both the positive and negative classes) as the evaluation criterion (Table S2).

Unlike the other classifiers, the blue band dynamic thresholding does not rely on labeled ground truth data. Instead, it uses orthomosaic-specific histogram characteristics to identify a unique threshold for each orthomosaic. A reflectance histogram was constructed from the field pixels' blue band using a bin size of 5, then smoothed using a centered moving average filter with a window width of five bins to suppress noise and enhance prominent modes. Local minima in the

smoothed histogram were identified and compared against a predefined threshold of either 127 or 90. Both thresholds have been previously used to distinguish snow-covered surfaces (Jacques-Hamilton et al., 2025; Salvatori et al 2011; Portenier et al 2020; Caparó Bellido and Rundquist 2021). The first local minimum exceeding this threshold was used to differentiate snow from no-snow pixels. In cases where no local minimum satisfied this condition, the predefined minimum

threshold was used.

## 2.4 SCA Model Performance Assessment

To evaluate the generalizability and robustness of the classifiers, four experiments were conducted. The first experiment (i.e., the base experiment) compares the classification performance for each classifier trained based on data from the 26 Phantom IV orthomosaics. The

Kingman Farm (750 sample points) and Thompson Farm (550 sample points) ground truth data were pooled and randomly split into 80% for training and 20% for testing. To account for the



variability introduced by random splitting, this process was repeated five times and the mean performance metrics was reported for each classifier. The remaining three experiments consider real-world challenges where instruments and locations vary, or data availability is limited.

The second experiment assessed the transferability of the classifiers between different optical sensors. The classifiers from the base experiment with the Phantom IV UAV camera imagery were used and their performances were assessed using the 10 orthomosaics (N = 500 sample points) from the Sony A5100 camera. Because dynamic thresholding determines classification thresholds independently for each orthomosaic and operates without labeled ground 315     truth data, this classifier fell outside the scope of Experiment 2.

The third experiment assessed the transferability of the classifiers between locations by training at one location and testing at another. This cross-site generalization experiment created new models for each of the ML classifiers and sites. Models were first trained using data from Kingman Farm and tested at Thompson Farm. A separate set of models were trained using the 320     Thompson Farm data then tested using the Kingman Farm data. Only the Phantom IV orthomosaics were used for this experiment. Also, because this experiment, like Experiment 2, presumes the use of labeled data, dynamic thresholding was not applicable.

The fourth and final experiment addressed flight planning to understand changes in performance based on the number of flights as well as the impacts of surface condition captured 325     during those flights. This experiment used subsets of orthomosaics to train new models, with subsets sizes ranging from 6 to 25 orthomosaics. Each subset was obtained by holding out $n$ orthomosaics from the original set of 26, thereby containing $26 - n$ orthomosaics. When the number of possible hold-out combinations was fewer than 500 (e.g., for $n = 1, 2$, or 3), all were used. Otherwise, 500 combinations were randomly sampled. For each combination, a new model




was trained and optimal hyperparameters were found following the methodology established in

the base experiment. Model performance was assessed on the corresponding held-out

orthomosaics (i.e., those excluded from training), hereafter referred to as held-out accuracy. Held-

out performances were summarized and reported for the range of training flights counts. This

experiment included only the top-performing classifiers from the base experiment for further

evaluation due to computational time needed to conduct this experiment.

## 2.5 Performance Metrics

The agreement between the ground truth data and the model output was quantified using

accuracy, balanced accuracy, F1 score, and Cohen Kappa performance metrics. In this study, a true

positive (TP) is defined as both the model and the ground truth agree on the "snow" class, while a

true negative (TN) indicates that both agree on the "no snow" class. A false positive (FP) occurs

when the ground truth dataset indicates "no snow" and the model output indicates "snow".

Similarly, a false negative (FN) occurs when the ground truth dataset indicates "snow" and the

model output indicates "no snow".

Accuracy is the number of cases where the model agrees with the ground truth data relative

to the total number of instances, calculated following Eq. (1)

$$Accuracy = \frac{(TP+TN)}{(TP+FP+TN+FN)} \tag{1}$$

Balanced accuracy is the average accuracy of both the positive and negative classes. It is

defined as the number of correct predictions for each class (true positives and true negatives)

divided by the total number of instances within that class, with the results averaged across all

classes calculated as



$$Balanced\ Accuracy = \frac{\frac{TP}{TP+FN}+\frac{TN}{TN+FN}}{2} \tag{2}$$

The F1 score is another widely used metric for evaluating classification performance, particularly for imbalanced datasets such as those having complete snow cover. The F1 score is the harmonic mean of precision and recall and is defined as

$$F1 = 2 \ \times \frac{Precision \times Recall}{Precision+Recall} \tag{3}$$

where precision is the ratio of correctly identified positive cases ("snow") to all predicted positives by UAS-SCA classifier

$$Precision = \ \frac{TP}{TP+FP} \tag{4}$$

and recall is the ratio of correctly identified positive cases to all actual positives in the ground truth

$$Recall = \ \frac{TP}{TP+FN} \tag{5}$$

Cohen Kappa is used to assess the reliability of a classifier while accounting for the possibility of agreement occurring by chance. It compares the observed agreement to the expected agreement under random classification and is defined as

$$K = \ \frac{p_0-p_e}{1-p_e} \tag{6}$$

where $p_o$, the relative observed agreement, and $p_e$, the expected agreement due to chance, are computed as

$$p_0 = \ \frac{TP+TN}{N} \tag{7}$$

$$p_e = \ \frac{(TP+FP)\times(TP+FN)}{N^2} + \ \frac{(TN+FN)\times(TN+FP)}{N^2} \tag{8}$$





where N is the total number of samples. A kappa value of 1 indicates perfect agreement, while a

value of 0 suggests agreement that is no better than that from random chance.

## 3 Results

### 3.1 Experiment 1: Baseline Performance

Experiment 1 (hereafter base experiment) evaluates the performance of seven classification

configurations for snow-covered area mapping with 26 orthomosaics using the P4 UAS platform.

The results clearly separated the tested configurations into two performance groups (Fig. 5). The

Maximum Likelihood, Random Forest, and SVM classifiers emerged as the top performers,

achieving accuracy, balanced accuracy, and F1 score values exceeding 0.95, indicating strong

agreement and classification accuracy. Their high performance was further supported by Cohen

kappa values exceeding 0.89. While SVM Blue Band and Static Blue Band classifiers consistently

underperformed the top three classifiers, a single blue band still yields reasonable results (balanced

accuracies of 0.86 and 0.84). While the Dynamic Blue Band classifier produced high accuracy

values for both thresholds (0.95 for a threshold of 90 and 0.93 for 127), the classifier had the

poorest performance for the other metrics. It should be noted that for the Dynamic Blue Band

classifier, the reported Cohen's Kappa represents an average across individual orthomosaics.

However, for orthomosaics containing only a single class (e.g., fully snow-covered or completely

snow-free scenes), Cohen's Kappa becomes unreliable, as it cannot account for agreement beyond

chance in the absence of class variability. Overall, all tested classification configurations have the

capacity to classify SCA with the Maximum Likelihood, Random Forest, and SVM being the most

robust classifiers.



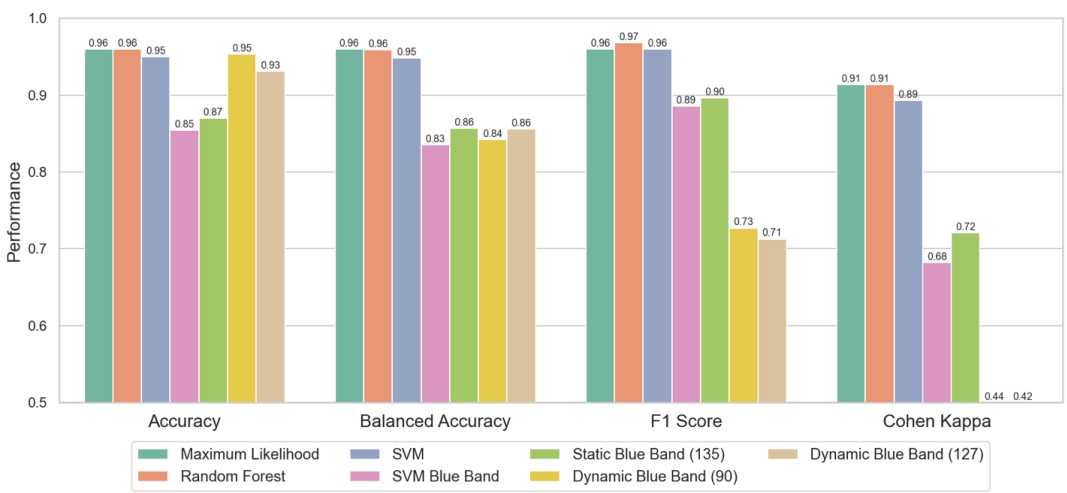


**Figure 5.** SCA performance results for base experiment 1 comparing observed SCA to modeled SCA by ML classification method and performance metric using 26 orthomosaics from Kingman Farm and Thompson Farm. Values below 0.5 for the dynamic blue band classifiers' Cohen Kappa
scores are listed but not graphed.

## 3.2 Experiment 2: Comparing Performance Across Cameras

Ten orthomosaics, five from each TF and KF, gathered with the Sony A5100 camera were

used to evaluate the models developed in the base experiment (except dynamic thresholding) using

the P4 orthomosaics. For this experiment, every classification configuration had accuracy and F1

score values exceeding 0.98 (Fig. 6), indicating no degradation in performance when transferring

from the P4 to the Sony camera. In fact, the models reported better performance values better when

tested on Sony orthomosaics compared to the base experiment. A closer examination of the Sony

orthomosaics reveals that this improvement is rooted in differences in snow conditions. The P4

dataset is much larger and contains a range of conditions including challenging conditions such as

heavy shadowing or illumination issues (e.g. Fig. 2 1/27/2025 and Fig. 3: 12/23/2020).

Coincidentally, the Sony orthomosaics were collected under conditions of relatively uniform snow





cover, limited shadow presence, and reduced illumination variability (see Fig. 4), making them more straightforward to classify. This is particularly evident for the blue band classifiers, whose accuracy performance improved from 0.86–0.88 to 0.97-0.98 with the exclusion of these challenging conditions. The Maximum Likelihood, Random Forest, and SVM classifiers' accuracy values, which exceeded 0.96 in the base model, rose by only 0.1-0.2.


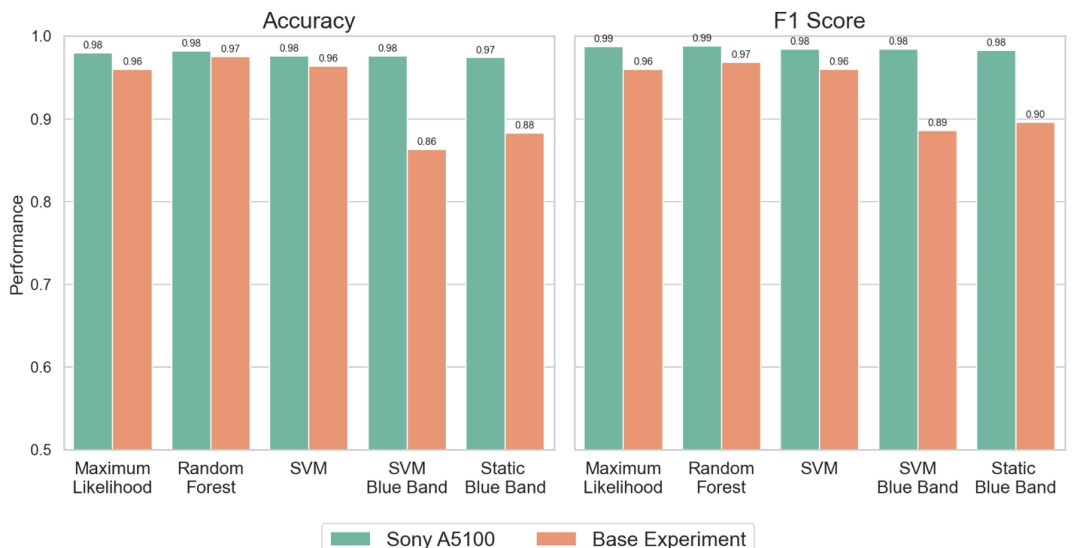

**Figure 6.** SCA accuracy and F1 score performance results for experiment 2 by ML classifier. SCA was modeled for the Sony A5100 orthomosaics using the ML models developed from Phantom IV (P4) orthomosaics. Base experiment results, ML models were trained and test using the 26 P4 orthomosaics, are shown for comparison.


**3.3 Experiment 3: Comparing Performance Across Sites**


Training and testing across different locations assesses the transferability of each classifier between sites. As in the base experiment, Maximum Likelihood, Random Forest, and SVM classifiers consistently had better transferability than both the SVM blue band and static blue band classifiers. The top three classifiers had performance metrics that typically exceeded 0.90. These





top-performing classifiers showed slightly better accuracies, 0.93 to 0.98, when trained on

Kingman Farm and tested on Thompson Farm, as compared to accuracy values of 0.91 to 0.94

when trained on Thompson Farm and tested on Kingman Farm. This may be due to the greater

variability of conditions at Kingman Farm, which trained the models for diverse snow conditions.

In contrast, the static blue band and SVM blue band classifiers continued to underperform. While

static blue band models yielded moderate balanced accuracies of 0.81, SVM blue band models

struggled significantly, with accuracy values of 0.58 and 0.78, underscoring its limited ability to

generalize between locations.

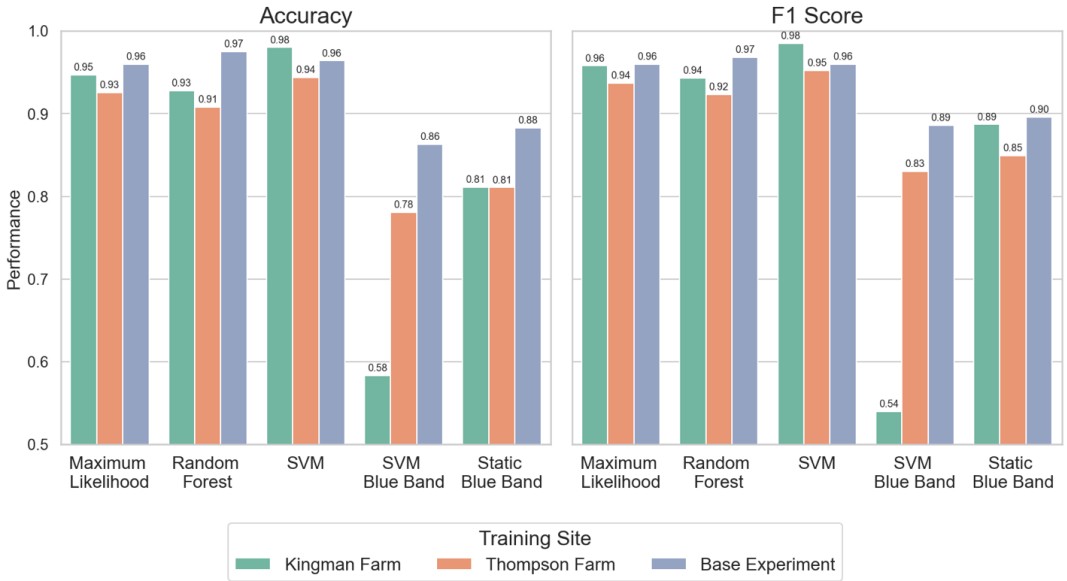

**Figure 7.** SCA accuracy and F1 score performance results for experiment 3 by ML classifier.
Modeled SCA performance when trained on either the Kingman Farm or the Thompson Farm site
and tested on the other site as compared to Phantom IV (P4) orthomosaics (26 images) base
experiment.



### 3.4 Experiment 4: Comparing Performance by Number of UAS Flights

Experiment 4 used subsets of 6 to 25 orthomosaics from the complete set of 26 orthomosaics to train and evaluate new MLE, SVM, and RF models (maximum 500 combinations). The performance was calculated for the individual orthomosaics that were "held-out" from the dataset, and an overall average of these held-out orthomosaics is presented here. Figure 8 shows that reducing the number of orthomosaics used for training only modestly reduces

the median combination's performance with mean accuracies remained consistently high (at or above 0.9) for all three classifiers. However, reducing the number of orthomosaics does lead to increased variability in performance among the combinations.

All three classifiers performed comparably when trained on larger datasets (> 20 orthomosaics) and had reduced ability on certain orthomosaics when trained on small datasets (<

12 orthomosaics). RF demonstrated the weakest generalization performance on mid-sized datasets. Even when most combinations performed well, certain combinations of training and testing datasets led to reduced performance. For models trained with 20 to 12 orthomosaics, fewer than 0.5% of MLE and SVM models had held-out accuracies below 90%, whereas approximately 4% of RF models fell below this threshold.

Even when the average of all held-out orthomosaics is high, some individual orthomosaics may have reduced performance (Fig. S-1). Among the three classifiers, the RF classifier is slightly more sensitive to training data availability. RF models' median held-out accuracies declined from 0.97 to 0.90 as the number of training orthomosaics decreased, whereas MLE and SVM only decreased from 0.97 to 0.95 and 0.98 to 0.94, respectively. The RF models also had individual

orthomosaic accuracies below 50% even when 18 orthomosaics were used for training (Fig. S-1). When the SVM and MLE training set included more than 14 orthomosaics, accuracy for nearly all





excluded orthomosaics exceeded 80% (Fig. S-1), though the SVM models had more low accuracy orthomosaics than MLE models. When the MLE and SVM training data sets contained about 10 or fewer training orthomosaics, some of the held-out orthomosaics had accuracies below 50%, 465    with a few extreme cases where accuracies were as low as 0%.





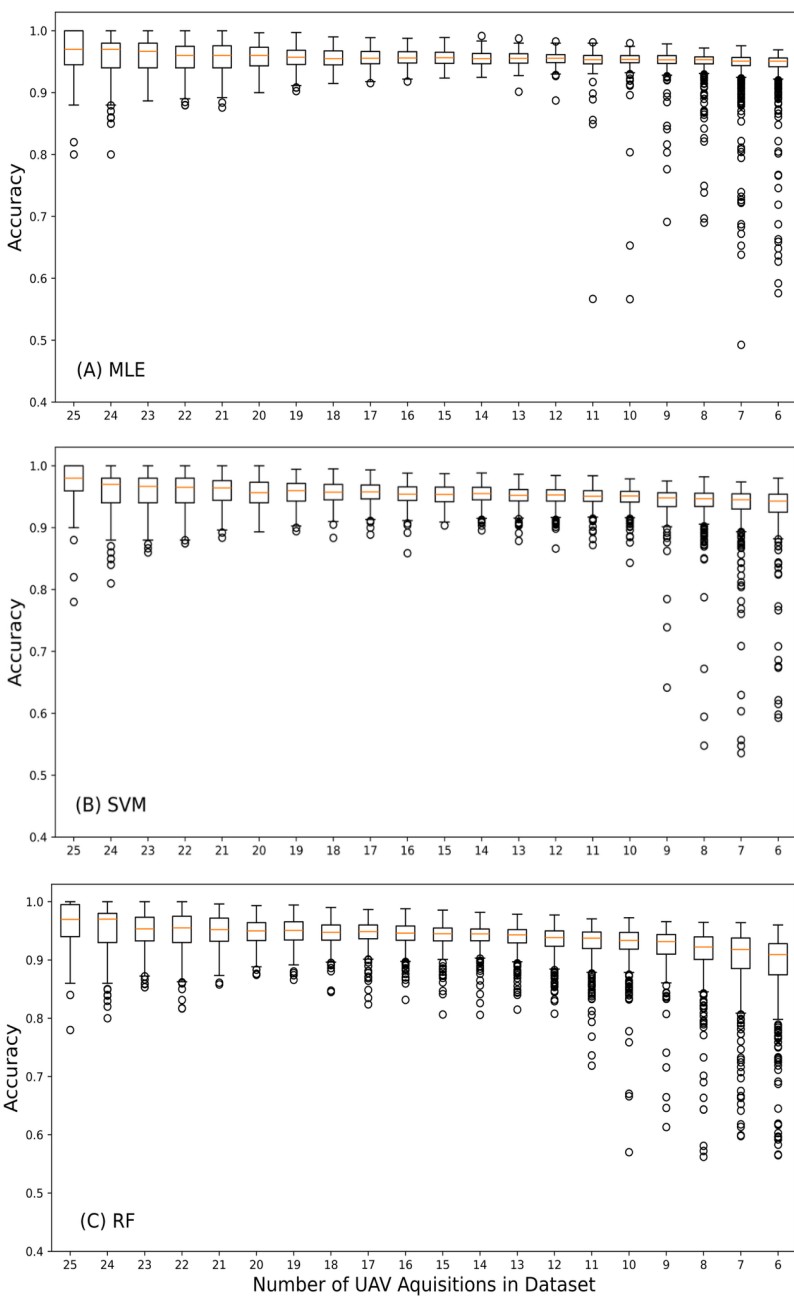

**Figure 8.** Boxplots of the mean held-out accuracy from each combination of held-out orthomosaics (maximum 500 combinations) by the number of orthomosaics included in the dataset for a) MLE, b) the SVM, and c) RF classifiers.






Individual orthomosaics' performance were examined. Figure 9 shows each individual orthomosaic's classification consistency when it was excluded from the training dataset plotted as the proportion of time that the classification accuracy was less than 80% by orthomosaic. While most scenes were reliably predicted with accuracies greater than 80% for more than 95% of the combinations, some scenes were consistently more difficult to classify. For the six lowest accuracy orthomosaics, the MLE models performed better than either the SVM or the random forest models. RF models have lower accuracies than either the MLE or SVM models except for the January 13th, 2025 orthomosaic from Kingman Farm. For example, the RF models' accuracies for the February 12th, 2024 orthomosaic of Kingman Farm were below 80% for nearly 40% of the combinations when that orthomosaic was held out from the training dataset, but for the same orthomosaic, MLE and SVM accuracies were seldom (10%) below 80%.







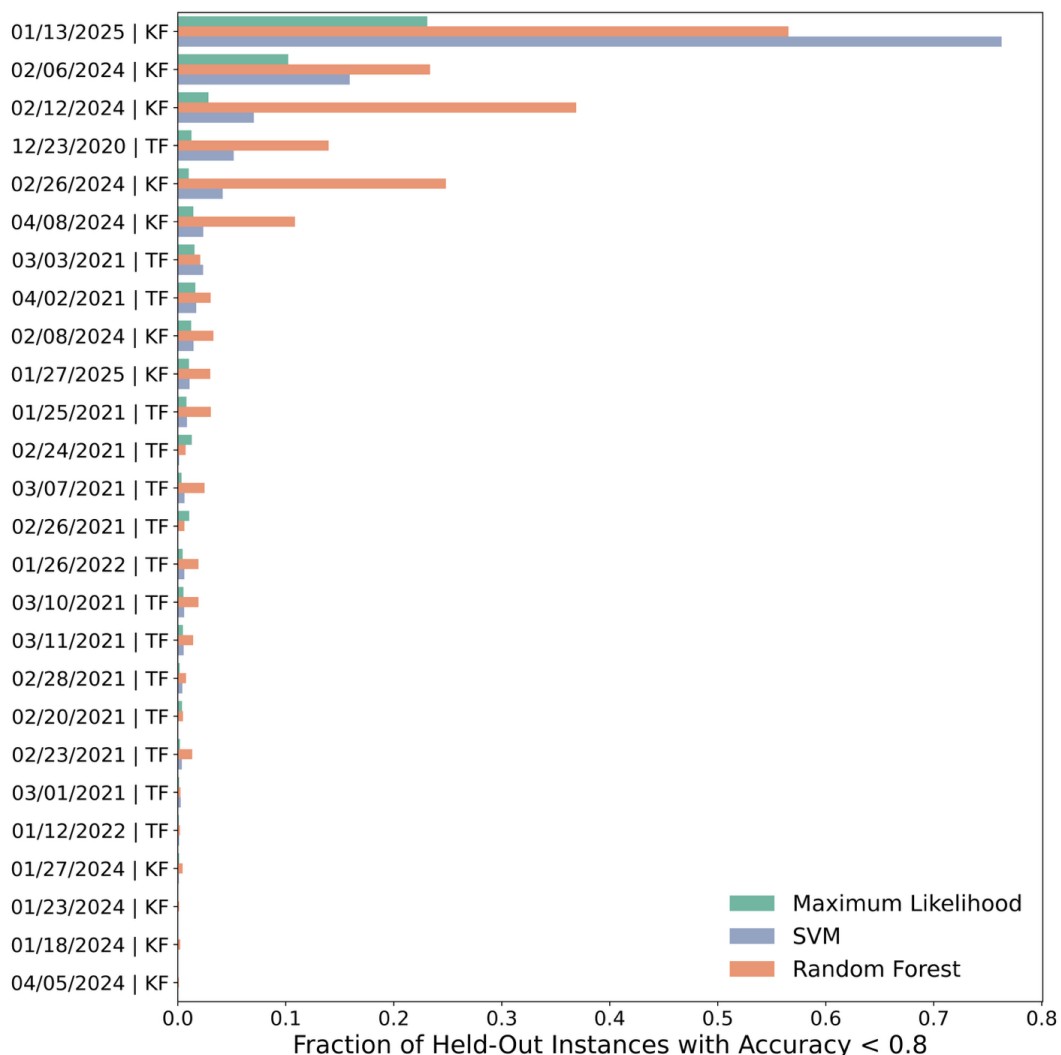

**Figure 9.** Fraction of held-out instances with classification accuracy below 80% for each date-site combination, shown separately for the three classifiers: Maximum Likelihood, Random Forest, and Support Vector Machine. Each bar represents the proportion of times a given orthomosaic was held-out and subsequently predicted with low accuracy.





To better understand which conditions led to reduced performance, we compared fractional snow-covered area (fSCA), snow wetness and cloud cover conditions for the ten highest- and

lowest-performing models for training datasets ranging from six to 12 orthomosaics.  fSCA for each orthomosaics was estimated using the models from the base experiment (i.e., models trained on the full dataset; see Fig. S-2). For each condition (i.e., fSCA, wetness, and cloud cover), we then compared its mean values from the dates included in the training sets with those of the corresponding held-out orthomosaics for the ten highest performing and up to 10 lowest-

performing when less 80% accuracy models.

For fSCA, the results for the SVM classifier are presented in Fig. 10, corresponding analyses for MLE and RF are shown in the supplementary materials (Fig. S-3 and S-4).  For datasets having ten to 12 orthomosaics, average fSCA ranges from about 0.5 to 0.8 for both the training and testing datasets. Note that this size training dataset has no models with less 80%

accuracy. As the dataset size decreases below ten, the highest-performing models still have fSCA values from 0.5 to 0.8, but the models having less 80% accuracy used training datasets with average fSCA values higher than 0.8. In addition, the held-out orthomosaics had lower fSCA values, sometimes dropping below 50%.  This reveals that the poorest performing models used training data are dominated by snow-covered orthomosaics (often >70% fSCA), then used to

predict conditions with lower or more variable snow coverage, Thus, there is a mismatch between the mean fSCA values of their training (included) and test (held-out) orthomosaics.  We did not find any evidence that differences in cloud cover (Fig. S-5, S-6, S-7) or snow wetness (Fig. S-8, S-9, S-10) between training and held-out orthomosaics influenced performance.




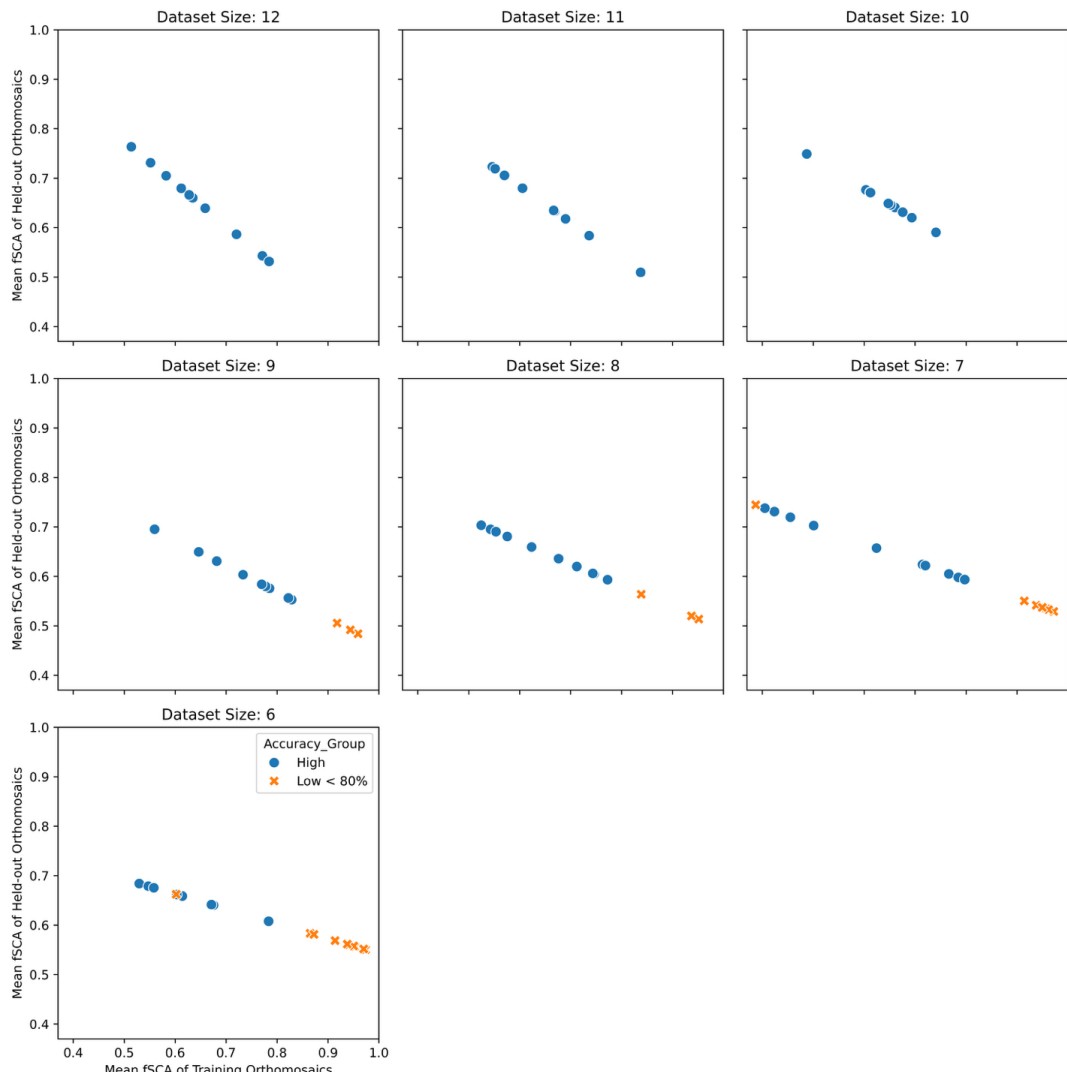

**Figure 10.** Mean fSCA of orthomosaics within training vs. Mean fSCA of held-out orthomosaics for the 20 combinations (10 highest and 10 lowest (<80%) accuracy) for the SVM classifier. Blue dots represent high-accuracy combinations; orange dots represent low-accuracy combinations. The individual plots differ by the number of orthomosaics in the training dataset. Training dataset sizes of 10, 11, and 12 did not have any held-out datasets with accuracy values < 80%.




## 4 Discussion

**4.1 Performance of UAS-based Snow-Covered Area Mapping Classifiers**

Our primary goal was to identify one or more classifiers that can rapidly, consistently, and accurately map SCA in open areas. Experiments 1 through 3 established that the three classifiers (MLE, Random Forest, and SVM) using full RGB inputs to be more effective classifiers for mapping SCA than classifiers relying solely on the blue band. In the first experiment, their ability

to differentiate between snow-on and snow-free conditions for 3 cm pixels exceeded 95% for accuracy, balanced accuracy, and F1 score performance metrics. In comparison to previous studies in open areas, this study's performance matches the performance reported by Eker et al.'s (2019) overall accuracy of 95% and exceeds Belmonte et al.'s (2021) accuracy of 90.2% using a random forest classifier, Cannistra et al.'s (2021) balanced accuracy of 0.82 using a convolutional neural

network-based method and Johnston et al.'s (2025) 80 to 88% accuracy via k-means clustering. Taken as a whole, the UAS performance appears to be similar to satellite snow cover products' performance which typically range from about 85 to 95% (Ault et al., 2006; Hall and Riggs 2007; Gao et al., 2010; Nolin, 2010; Frei et al., 2012; Bair et al., 2016; Coll and Li, 2018) with the UAS identifying SCA at a much finer resolution (cm scale). The RGB classifiers also demonstrated

stable performance when applied to imagery acquired from a different optical sensor. However, we note that in this study, the RGB intensity values from both sensors spanned a similar range (i.e., 0 to 255; Fig. S-11). In cases where a sensor exhibits reduced sensitivity in a spectral channel or produces compressed or biased RGB values, it is critical to apply bias correction to the intensity values prior to classification to ensure comparability and preserve model performance.



540        Classifiers relying solely on the blue band, such as SVM Blue Band, Static Blue Band

Thresholding, and Dynamic Blue Band Thresholding, still achieved good results, but were

consistently less accurate than their RGB-based counterparts. Of the blue band classifiers, Static

Blue Band Thresholding performed the best based on its F1 Score of 0.90, balanced accuracy of

0.86 and accuracy of 0.87. While the Dynamic Blue Band Thresholding had an accuracy (0.95)

comparable to the MLE, RF, and SVM, this is largely due to its strength in capturing snow when

fSCA is high, its balanced accuracy of 0.78 reflects its diminished ability to capture mixed

conditions. The dynamic thresholding approach generally underestimates snow-covered pixels, as

indicated by a higher rate of incorrectly classifying of snow free pixels than snow covered pixels

(Tables S-3 and S-4). While this classifier is designed to adapt to varying illumination conditions

unique to each orthomosaic, its effectiveness diminishes when no local minimum is detected in the

histogram (e.g., fully snow-covered days) making its performance highly dependent on the pre-

defined minimum threshold. Figures S-12 and S-13 clearly demonstrate how the value of the pre-

defined minimum threshold can influence classifier performance when the intensity histogram

lacks a clear local minimum. Moreover, the dynamic thresholding approach can be memory-

intensive when constructing histograms using all pixels from high-resolution orthomosaics for

large study areas. As an alternative, random sampling of the orthomosaic pixels may be used to

reduce memory requirements but would likely further reduce classification performance due to

less representative histogram estimation. Despite its simplicity, the Static Blue Band classifier is

preferable to the dynamic approach. Where reduced performance is an acceptable trade-off for

increased ease of use and users are knowledgeable about conditions that may be problematic, the

blue band threshold classifier appears to offer a reasonable compromise that merits additional

vetting.



The clear difference between using classifiers with three bands as compared to the single blue-channel is somewhat unexpected because the blue-channel is a widely used classifier. As reviewed by Webster & Jonas (2018), the use of the single blue-channel is the favored method for RGB imagery from field cameras because blue light is reduced in the canopy view fraction (e.g. Reid and Essery, 2013; Frazer et al., 1999) and saturates over most snow-covered scenes causing loss of sun-lit/shadow contrast (Wolter et al., 2012; Dozier, 1989). Using UAS RGB imagery, Jacques-Hamilton et al. (2025) reporting satisfactory results by thresholding the blue-channel values following Salvatori et al.'s (2011) approach for mapping snow cover using field camera images. Their blue pixel intensity threshold value of 90 was determined using a qualitative approach in which they adjusted the threshold until pixels classified as snow cover agreed with what they saw in their photos. This is considerably lower than the threshold value of 127 used for field cameras (e.g. Salvatori et al 2011, Portenier et al 2020, Caparó Bellido and Rundquist 2021).

The importance of using all three RGB bands is also apparent when training and testing across different locations (Experiment 3). Although both SVM using the RGB band and SVM Blue Band used the SVM classifier, the RGB version maintained high accuracy across all training sites (always above 94%), while the blue-band-only version (SVM Blue Band) consistently had the lowest observed accuracies, dropping as low as 58%. This suggests that while the blue band is highly sensitive to snow due to its strong reflectance properties (Nolin, 2010) combining it with the red and green bands provides more reliable snow classification. Our results also indicate that incorporating full RGB inputs is particularly valuable when mapping shadowed snow, dirty snow, and wet snow. Figure 11 shows examples of SCA maps on a partially snow-covered day (Fig. 11A) and a fully snow-covered day (Fig. 11B). It appears that the reduced blue band reflectance caused by shadows, impurities, or melting led to misclassification of snow-covered surfaces leading to a



reduction in the field's fSCA values by 15 to 25% in comparison to fSCA that was properly captured by MLE, Random Forest, and SVM.

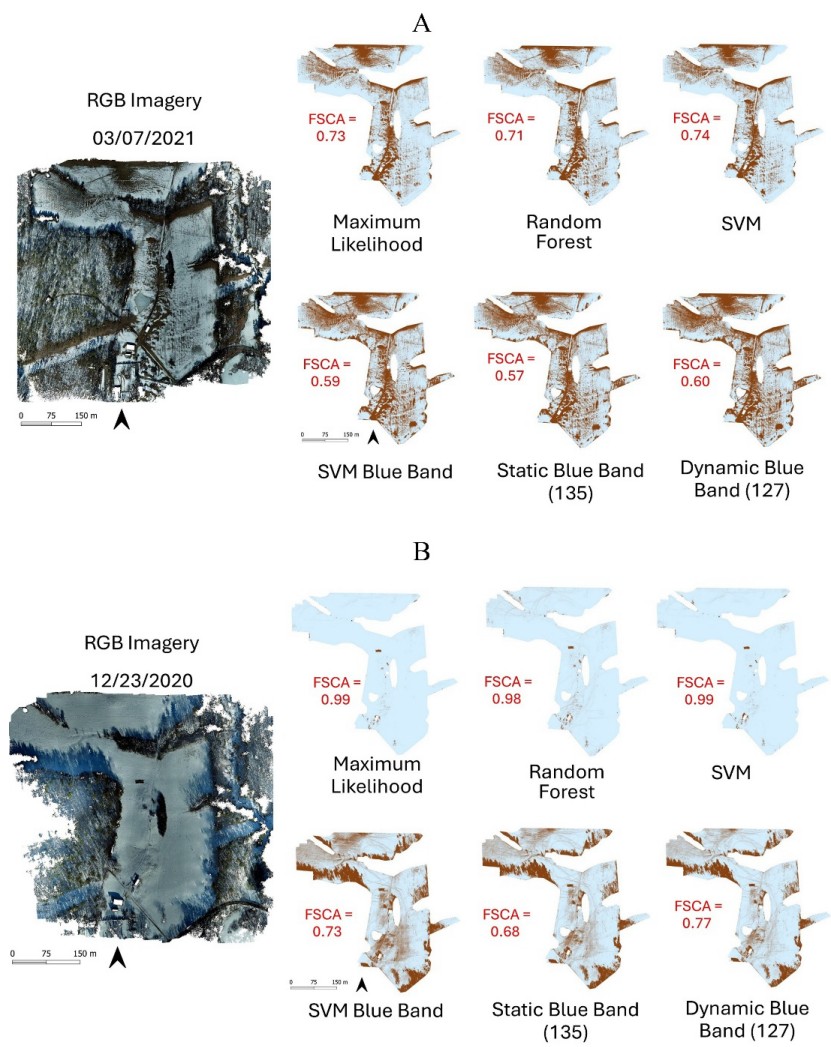

**Figure 11.** UAS orthomosaics from Thompson Farm on A) March 7[th], 2021 and B) December 23[rd], 2020 derived and SCA classification using each of the six classifiers. Fractional snow cover (fSCA) is noted for each classifier. Purple circles delineate areas of deep shade on December 23[rd], 2020.

590



**4.2 Recommendation for methods, transferability, and UAS data collection strategies**

Experiments 1 through 3 consistently identified MLE, Random Forest, and SVM as the most effective classifiers for mapping SCA using UAS-derived RGB imagery. Across all three experiments, these classifiers produced comparable results in terms of accuracy, F1 scores, and other performance metrics. Our assessment of model generalizability found surprising high performance for these SCA models when trained at one location and tested at another. Breen et al.'s (2024) neural network model, trained to capture snow depth from field camera imagery for Colorado sites, required the model to be fine-tuned using images from the new sites in Washington State to achieve acceptable performance. Although our Thompson Farm and Kingman Farm sites were separated by less than 10 km, the two sites' orthomosaics were from different years with different snow conditions. Model generalizability between the widely used P4 camera and the Sony A5100 camera appears to be even less critical. The results of experiment 4 do suggest that RF models may be more sensitive to some surface conditions, possibly due to its reliance on multiple decision trees that are impacted by the absence of key features in outlier orthomosaics. Conversely, SVM and MLE demonstrated relatively more stable behavior, indicating that they may be less prone to overfitting and could offer better generalization for snow detection when the training set is not representative of a new orthomosaic's conditions. MLE, while simple, seems to perform consistently in the majority of cases.

Our generalization experiments also offered insights into how both the number and characteristics of orthomosaics influence the development of a robust model, highlighting the challenges posed by data availability and diverse snow cover conditions when mapping snow using UAS RGB imagery and machine learning classifiers. For dynamic winter conditions similar to those in our study area (characterized by an ephemeral snowpack) at least 12 orthomosaics





spanning a range of snow cover conditions was enough to result it a reasonable classification accuracy (exceeding 80%) when employing robust classifiers such as MLE, SVM, and RF. However, beyond just the number of orthomosaics, such a dataset should be representative of a diverse range of snow cover conditions that pose challenges for snow mapping. Key examples of

challenging conditions include:

*Thin snow*: The two orthomosaics with the highest frequency of accuracies below 80% (January 13th, 2025, *KF* and February 6th, 2024, *KF*) in Experiment 4 also exhibited the lowest median accuracy across all models (Fig. S-14). Although both orthomosaics were predominantly snow-covered, with fSCA exceeding 60%, the observed snow depth was relatively shallow (3 to 4

cm). This consistently poor generalization performance, regardless of classifier, underscores the inherent difficulty of detecting shallow snow cover using only optical imagery. These findings are not surprising for these optically thin snowpacks. In the visible wavelengths, where there is little absorption and scattering is dominant, light penetrates snow by 10's of centimeters beneath the surface (deeper for blue than red). If snow is shallower than this then almost certainly the observed

signal is being darkened by the ground below. In contrast, methods that use the NIR, where ice absorption/grain size dominates, would be expected to perform better because light does not penetrate below 1-2 cm of snow. That said, our experiment used numerous other orthomosaics taken when snowpacks are equally thin. In most cases, these thin snowpacks were successfully classified, suggesting that even if the ground is making the snow appear darker than it actually is,

it still is brighter than most other surface types in this mapping area. Special cases such as shallow and wet snow or snow that is shallow and shadowed require additional study (personnel communication S. McKenzie Skiles, March 19, 2025).



*Patchy snow:* The Experiment 4 results suggest that generalization failure may be rooted

in an fSCA mismatch between training and test sets. When classifiers are not exposed to a diverse

range of snow cover conditions, their ability to identify snow accurately in less homogeneous

scenes is significantly impaired. This vulnerability is particularly evident for smaller training sizes,

where the representation of snow variability decreases. Patchy snow scenes often contain a mix of

exposed ground, water-stained snow, and potentially dirty snow with soil or organic debris. These

features cause snow RGB values to deviate from values for fresher and cleaner snow typically

represented in training datasets with higher mean fSCA. These findings align with previous studies

in snow classification and remote sensing, which have shown that snow detection algorithms,

particularly those relying solely on RGB or optical inputs, are sensitive to scene complexity and

snow spatial heterogeneity (Painter et al., 2009; Dietz et al., 2012). Our results further show that

this sensitivity is not limited to spectral thresholds or empirical rules but extends to machine

learning classifiers as well. Both random forest and SVM rely on feature-space separation, and if

the features of aged or dirty snow overlap too much with non-snow classes, the models may

misclassify them unless those cases were explicitly represented during training. MLE, in turn,

assumes underlying distributional properties for each class. When snow pixels in patchy conditions

no longer follow the same statistical profile learned from clean snow scenes, misclassification

becomes inevitable. Previous studies have also found that snow reflectance varies with solar

illumination, becoming brighter and more neutral under direct sunlight and darker and bluer in

shadow or diffuse light (Dozier, 1989; Painter et al., 2003). However, our results did not find any



evidence that variability in weather conditions at the time of UAS acquisition limited

generalization performance.

*No snow*: Three additional orthomosaics with a high fraction of low-accuracy outcomes in

Experiment 4 (*February 12th, 2024, KF*, *February 26t, KF, 2024*, and *April 8th, 2024 KF*) further

illustrate the limitations of relying solely on RGB data for snow mapping. These cases, all from

the Kingman Farm site, feature bare ground with a whitish appearance that can be easily

misclassified as snow (Fig. S-15). This situation is likely not a problem if the same research team

collects the imagery and processes it because the lack of snow is self-evident. If a different team

processes imagery, this issue can be addressed by a manually scanning the orthomosaics prior to

classifying snow.

From a practical standpoint, insights from this study can guide future data collection,

dataset design, and model development. It is essential to represent the full spectrum of snow depth

and fSCA from bare ground to full snow cover, and especially the transitional states (e.g., fSCA

between 20–60%). While not measured in this study, representativeness likely extends to a range

of snow metamorphism stages. As snow ages, grain size growth and increased density reduce its

reflectance, particularly in visible bands, leading to a duller appearance in RGB images (Warren,

1982; Green et al., 2006). Wet, refrozen, or contaminated snow further lowers reflectance, making

it spectrally similar to bright bare ground, introducing noise and complexity (Kaufman et al.,

2002). These effects result in a strongly non-linear relationship between fSCA and observed snow's

spectral characteristics, even in the limited RGB space.



## 5 Conclusion


Mapping snow cover persistence via UASs is useful for detailing snow spatial variability over large areas, and has direct relevance to agriculture, hydrology, mobility, and land surface modeling. This work compares seven classification configurations for producing snow cover estimates from UAS's high resolution (5 cm) observations over two study sites located in southern

New Hampshire, USA having open, agricultural landscapes. It further offers guidance on effective strategies for producing UAS derived SCA maps for a range of purposes such as validation of satellite snow cover products, converting satellite NDSI observations to fSCA, and downscaling satellite SCA observations. The tested classifiers included machine learning methods (Random Forest, Support Vector Machine, and Maximum Likelihood Estimation) and thresholding

approaches (static and dynamic). Predictor selection varied by classifier, using either full RGB intensity values or the blue band alone. Among them, MLE, RF, and SVM using RGB inputs consistently outperformed others, with accuracy, balanced accuracy, and F1 scores exceeding 0.95. These classifiers also demonstrated strong generalization across sensors, locations, and winters. In contrast, classifiers relying solely on the blue band showed lower overall performance (accuracy:

0.90; balanced accuracy: 0.85; F1 score: 0.81, on average) and reduced transferability. For snow mapping using UAS optical imagery in regions characterized by shallow, ephemeral snow and dynamic winter conditions, we found that including at least 12 orthomosaics spanning a broad range of snow conditions, especially including transitional states (e.g., fractional snow-covered area between 20–60% during melting, reaccumulation and aging phases) can yield accuracies

above 80% with MLE, SVM, and RF. While MLE, SVM, and RF performed comparably when trained on large, representative datasets, MLE and SVM emerged as more reliable and consistent

choices when data availability is limited. Where reduced performance is acceptable, the blue band threshold classifier appears to offer an easy-to-use method, but users are advised to carefully vet their dataset for conditions that may be problematic.

## Data Availability

Meteorological observations used in this study were obtained from United States Climate Reference Network (USCRN) stations and are publicly available from the NOAA National Centers for Environmental Information (NCEI) at https://www.ncei.noaa.gov/access/crn/.

## Author Contributions

Methodology: MM, AF, AH, JMJ; Data curation: AF, AH, MM; Software: MM; Formal analysis: MM, AF; Visualization: MM, AF; Writing (original draft preparation): MM, AF, AH, JMJ; Writing (review and editing): MM, JMJ

## Competing Interests

The authors declare that they have no competing interests.

## Acknowledgments

The authors want to thank Jeremy Johnston and Rashik Koirala who participated in the data collection which made this work possible. We also thank Lee Friess for editing this manuscript. This material is based upon work supported by the Broad Agency Announcement Program and the Cold Regions Research and Engineering Laboratory (ERDC-CRREL) under Contracts W913E521C0006 and W913E523C0004. Distribution A: Approved for public release. Distribution is unlimited.



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
