# Peer review of "Identifying Snow-Covered Areas from Unoccupied Aerial Systems (UAS) Visible Imagery: A Comparison of Methods"

_EGUsphere, 2025_

## Referee Comment (RC1)

**Identifying Snow-Covered Areas from Unoccupied Aerial Systems (UAS) Visible Imagery: A Comparison of Methods**

**By Moradi et al. (2025)**

This manuscript compares several different semi-automated classifiers for mapping snow covered area from UAS RGB images across two study sites in New Hampshire, USA. UAS data were collected several times over several winters using two different sensors. The authors use this rich dataset to conduct multiple experiments. In general, the paper is well written, but would benefit by including additional work in their defined experiments. As is, I am not convinced that the methodological framework and conclusions in the current manuscript are a substantial improvement to our greater understanding of SCA derived from UAS RGB images. With that said, I believe that major revisions are necessary before consideration for publication.

**Major Comments:**

Overall the principle objectives are solid and worth pursuing. However, the objectives can be addressed more thoroughly with a few additions or revisions:

Throughout various sections of the paper the authors imply several relationships or causal drivers influencing model performance but never present formal statistical testing such as an analysis of variance, t-test, or regression. Without the accompanying metrics and indicating significant differences, these relationships should be reframed as qualitative observations.

**Experiment 1: Baseline performance**

Overall the section is sound in its presentation of model performance metrics, although can be much improved. A major limitation in this experiment, which cascades down into the subsequent is that the authors classify images into 2 classes, snow and bare ground. Admittingly, we are only interested in snow covered area, however we are also interested evaluating methods, which means that we are interested in how the models agree, but I am more interested in how, where, and why the models disagree. And to get that at those distinctions, the UAS imagery needs to be classified into more classes like shadow, snow, vegetation, bare ground.

Without such classifications it is challenging to suggest with confidence which classifier might perform best under shadowy conditions etc.

Further, this study has a valuable opportunity to incorporate lidar. Lidar can provide an independent dataset for mapping SCA. As well as help classify snow types which can further be used to aggregate model performance.

**Experiment 2: Cross-sensor Comparison**

Though relevant, I believe as the experiment is currently setup this is not only a cross-sensor but also a cross-date comparison. In order to better isolate the differences model performance across sensors, if the model is trained using images from camera A, then it needs to be tested using

images co-collected imagery from A and B. This allows us to distinguish model performance relative to camera A.

Experiment 3: Across Sites

Similar to experiment 2, I think working with UAS imagery the day and time of day of acquisition is so important. In this experiment, it is again difficult to parse out what parts of model performance are being affected by location or day of acquisition?

Results for experiment 2 and 3 were similar in that differences in model performance were not from sensor or location but rather differences in date of acquisition. It would be ideal if both experiments had isolated the effect of acquisition date.

Reviewing these three experiments I would suggest excluding the sensor comparison and even only focus on the images acquired from the A5100. Even though that reduces the total number of acquisitions from 36 to 10. I believe the reduction in imagery is worth the ability to incorporate lidar into the study, which will allow for the introduction of an independent snow cover map as well as aggregating snow cover classifications by snow depth.

**Specific Comments:**

The current figures are a bit repetitive and difficult to read mainly because there is little variability in the performance metrics. These data and model outputs can be presented in several interesting ways including aggregated area values as timeseries or more qualitative approaches that identify model outputs misclassed as shadow, for example.

Random forest is spelled out in several places. It should be consistently presented as 'RF.'

In the sampling approach, how were shadows excluded form the random sample? It seems it could be possible that a randomly selected pixel could be a shadow? Were classes equally distributed, 25 for snow and 25 for bare ground?

In the discussion 4.1, it states 'Our primary goal was to identify one or more classifiers that can rapidly...' Rapidly implies that model performance speed was tested, no metrics are presented but could be valuable.